# Experimental Study on the Interrelationship between the Moisture Content and Drying Shrinkage of Autoclaved Aerated Concrete Wallboard

**DOI:** 10.3390/ma15165582

**Published:** 2022-08-15

**Authors:** Jianyu Yang, Jiaming Zou, Weijun Yang, Sirong Yi, Jinzhao Liu

**Affiliations:** School of Civil Engineering, Changsha University of Science and Technology, Changsha 410014, China

**Keywords:** autoclaved aerated concrete, wallboard, moisture content, drying shrinkage, equilibrium moisture content, drying shrinkage value, experimental study

## Abstract

Autoclaved aerated concrete wallboard (AACW) has been widely used as a building envelope component in the infill walls of frame structures, which has broad prospects for development and utilization. However, the cracking of AACW has become a pressing problem, and this problem needs be solved or relieved effectively. We need an effective control method to reduce the cracking problem of AACW. It is necessary to study the interrelationship between the moisture content and the dry shrinkage of AACW. In this paper, a moisture content test and a drying shrinkage test of AACW were conducted, to understand the effect of the moisture content on the drying shrinkage performance of AACW. In addition, the moisture content of AACW with time was explored, and changes in the dry shrinkage value of AACW with the moisture content of AACW were obtained. According to the results and the conditions and the hypothesis of the test, the drying shrinkage value of AACW increases with time, and the drying shrinkage speed was fast in the early stage and tended to be stable in the later stage. In AACW, the drying shrinkage value and the relative humidity have a notable negative correlation. In addition, there was a positive correlation between the drying shrinkage value and the initial moisture content and the ambient temperature. When the AACW lost water from its initial moisture content to the equilibrium moisture content, the accumulated dry shrinkage value of AACW increased with the water loss. Moreover, a time-varying model of the moisture content and a prediction model of the equilibrium moisture content of AACW were established, and time-varying models of the drying shrinkage value of AACW with different initial moisture contents were proposed. The results provide a scientific basis for the reasonable maintenance and profitable control of drying shrinkage cracking of AACW.

## 1. Introduction

Autoclaved aerated concrete wallboard (AACW) is widely used worldwide as a new type of material. However, due to its low strength, high hygroscopicity, relatively large drying shrinkage value, and fairly low crack resistance, some quality problems with AACW often occur in the process of its promotion and use, such as wall cracking, plaster layer bulging, leakage, etc., affecting the quality of construction and the use of AACW. Therefore, the cracking of AACW has become a pressing problem, and this problem needs be solved or relieved effectively. Moreover, we need an effective control method, to reduce the cracking problem of AACW.

For autoclaved aerated concrete products, the moisture content at the time of forming the wall has a significant impact on its properties, such as the strength and the drying shrinkage. These properties were essential parameters for many theoretical analyses and engineering applications. *Autoclaved Aerated Concrete-Properties and Structural Design*, published in 2005 in the United States, takes the thermal conductivity of autoclaved aerated concrete at 5% moisture content as its theoretical calculation value [1]. Chinese specifications specify a 3% moisture content by volume as the equilibrium moisture content of autoclaved aerated concrete of different bulk densities [2]. Abdou, Campanale, et al. [3,4] found that the higher the temperature and the higher the moisture content, the higher the thermal conductivity. Moreover, Fan and Li-Wu discussed the relationship between moisture content and autoclaved aerated concrete and proposed a fractal model to predict the effective thermal conductivity [5]. The setting of moisture content in the specification also mainly considers the influence of moisture content thermal performance on the wall [6]. There is a lack of research on the effects of the moisture content of wallboard on drying shrinkage, while there are some studies on the moisture content of brick products on walls, such as a long-term drying shrinkage test study on autoclaved fly ash bricks, which showed that [7] dry shrinkage increases with the increase of moisture content of the upper wall. The maximum dry shrinkage value of watering bricks (11%) in wet, medium, and dry environments was 3.5-, 4.1-, and 2.5-times that of the non-watering bricks (2.9%), respectively. In addition, controlling the moisture content of the wallboard and other products had a vital role in reducing the wall drying shrinkage and crack deformation.

Therefore, the most important factor is the moisture content of AACW, which affects the drying shrinkage value of AACW. However, the relationship between the drying shrinkage value and the moisture content of AACW has not been revealed and understood. In order to describe this relationship clearly, some definitions are explained, as follows: When the AACW loses water, from the initial moisture content to the equilibrium moisture content, the unit length shrinkage value of AACW in this process is named the drying shrinkage value. The equilibrium moisture content means the final stable moisture content in a particular air state (temperature, relative humidity). Moreover, initial moisture content means the moisture content of AACW at the point of installation and construction. It can be seen that there is a close relationship between the material properties, maintenance, environmental temperature, humidity, etc. Accordingly, an effective controlling method to control the cracks for reducing the drying shrinkage and crack deformation of AACW should be researched. Moreover, we need to explore the relationship between the drying shrinkage value and moisture content, especially regarding the equilibrium moisture content and the initial moisture content. Therefore, these relationships are in favor of acquiring a controlling method and are helpful for research.

In this paper, the cracking problem of AACW was investigated from the perspective of the effect of moisture content on drying shrinkage. Furthermore, a moisture content test and a drying shrinkage test of AACW were conducted, to understand the effect of the moisture content on the drying shrinkage performance of AACW. In addition, we hoped to obtain the time-varying law of the moisture content and a prediction model of the equilibrium moisture content of AACW and the time-varying relationship of the drying shrinkage value of AACW with different initial moisture contents. These results could provided a scientific basis for wallboard maintenance and drying shrinkage crack control.

## 2. Moisture Content Test of AACW

### 2.1. Selection and Test Scheme

From AACW just out of the kettle were taken 15 surfaces without honeycombing, cracks, pores, and other defects of the specimen (Figure 1); specimen size 400 × 100 × 100 mm. The main instruments and equipment used in the test were a standard constant temperature, humidity curing box, micrometer, electric blast drying box, electronic weighing, electronic temperature and humidity meter, and soaking box.

The test with an electric thermostatic blast dryer set 5 different temperatures, to create 5 different test environments. The test block was saturated with water in different test environments, and the the moisture content of the test block was tested every day, until the test block was down to the equilibrium moisture content. The test program is shown in Table 1.

The specific test steps were as follows:(1)Soak all the missed water (20 °C ± 2 °C) for 72 h;(2)Take out the test block from the water, put it on the shelf for 1 min, then wipe off the water from the surface of the test block with a dry towel, and then weigh the initial mass m0 of each test block;(3)Set the drying oven temperature according to Table 1, and put the corresponding test piece groups in different test environments. Measure the temperature T, relative humidity RH, and the mass mt of the test block every 24 h. When the mass fluctuation of the test block before and after the two measurements is less than 0.01 kg, it can be determined that it has reached an equilibrium moisture content, and the mass of the test block at this moment can be measured;(4)Put the test block into the box with a temperature of 105 °C for drying. When the mass difference before and after the test block is not more than 0.2%, measure the weight m_d_ of the test piece after drying;(5)Moisture content of the test block at time t under different test conditions:
(1)$$\omega_{t}=\frac{m_{t}-m_{d}}{m_{d}}\times 100\%$$

In this formula, *ω**_t_*** is the moisture content (%) of the test block at time *t*; *m_t_* is the mass of the test block at a time (kg); *m_d_* is the mass of the test block when it is absolutely dry (kg).

The average moisture content of the three test blocks under the same test environment is taken as the moisture content of the wallboard under the test environment.

(6)The final balanced moisture content of the test block under different test environments [8]:
(2)$$\omega_{e}=\frac{m_{e}-m_{d}}{m_{d}}\times 100\%$$

In this formula, *ω**_e_*** is the final equilibrium moisture content (%) of the test block; *m**_e_*** is the mass of the test block at the equilibrium moisture content (kg); *m**_d_*** is the mass of the test block when it is scorched (kg). The average of the equilibrium moisture content of the three test blocks under the same test environment is taken as the final equilibrium moisture content of the wallboard under the test environment.

### 2.2. Test Results and Analysis

The daily ambient temperature and corresponding relative humidity of each group of test blocks are shown in Table 2.

The test measured the change of moisture content of AACW from the kettle to the equilibrium moisture content time; Table 3 records the daily moisture content of the five groups of wallboard specimens under different ambient temperatures and relative humidity, and the moisture content of 0 d in Table 3 is the moisture content of the specimens just placed in the box after being saturated with water. From the test results in Table 3, the moisture content curve of the specimens under different test environments with time was obtained, as shown in Figure 2.

From Table 3 and Figure 2:(1)The moisture content of the AACW specimens was measured to be as high as 69.6~71.8% in the saturated state of water. This is because autoclaved aerated concrete is a macroporous material, the total porosity of which is measured by the mercury piezoelectric method to be as high as 76% or more; SEM imaging shows that most of its pores are through pores, and the macro pores are in the shape of ink bottles with a “small mouth and big belly” [9]. The pore characteristics of autoclaved aerated concrete enable it to adsorb more water in the immersed state, so that the wallboard specimens show the characteristics of a high saturation moisture content.(2)The equilibrium moisture content of the AACW specimens in groups 1 to 5 in the test was 5.1%, 2.0%, 1.3%, 1.0%, and 0.8%, in order, which shows that the higher the ambient temperature and lower the relative humidity of the wallboard specimens, the lower the equilibrium moisture content. This is because, according to the Kelvin formula, under any relative humidity, only the water in holes more significant than the Kelvin radius will evaporate. The lower the relative humidity, the smaller the critical radius of water evaporation, the greater the water loss of the material, and the lower its equilibrium moisture content.(3)In the early stages, AACW specimens’ moisture content decreases rapidly, then this decreases gradually to zero, and finally, the specimen reaches the equilibrium moisture content. This is because the water in the specimen saturated with water is mainly free water, capillary water, adsorbed water, interlayer water, and combined water.(4)The higher the temperature and lower the relative humidity of the test environment in which the AACW specimen is located, the less time it takes to move from the saturated moisture content to the equilibrium moisture content. This is because the surface of the AACW saturated with water contains a large amount of free water. At this moment, the drying process is controlled by the rate of vaporization of surface water and the rate of vaporization of water on the surface of the specimen is affected by the temperature and humidity of the test environment. The limit value of moisture that the air can retain at different temperatures decreases as the air temperature rises, and when the ambient temperature rises, the ability of dry air to absorb moisture will increase. The energy of water molecules in the material will also increase; thus, the higher the ambient temperature, the faster the rate of water loss in the test piece. According to Fick’s law, in the molecular diffusion process, the lower the concentration of molecules in the surrounding environment, the faster the rate of molecular diffusion; thus, the lower the relative humidity of the environment, the faster the evaporation of water from the specimen. Therefore, the influence of ambient temperature and relative humidity on the rate of moisture loss and the time required to reach equilibrium in AACW tests is reasonable.

## 3. Time-Varying Model of Moisture Content

Since the moisture content of the AACW was very high when they came out of the kettle, they were also quickly in a high moisture content state after being soaked by rain or water. Therefore, the wallboard must be maintain an acceptable moisture content before installation and use. Otherwise, a high moisture content will easily cause the AACW to crack. The time-varying model of the moisture content of wallboard can provide a theoretical basis for the time limit of wallboard installation.

Empirical models, semi-empirical models, or theoretical models are usually used to simulate the change of material moisture ratio with time during desorption, and the most widely used mathematical models of drying kinetics are shown in Table 4 [10,11,12,13].

The moisture content in Table 3 was converted to moisture ratio, based on the model in Table 4. AACW drying kinetic models under different test environments were fitted, and the model parameters are shown in Table 5. The decidable coefficients *R*^2^ fitted to each model are shown in Table 6. A comparison between the measured values of the moisture ratio of AACW under five test environments with time and the fitted model curves is shown in Figure 3.

From Table 6 and Figure 3, the fitted model curves based on the Lewis, Page, and Henderson–Pabis models deviate less from the measured values, and the fitting effect is better. The decidability coefficients *R*^2^ of these three fitted modes were above 0.97. The Lewis model is simple in form, so the Lewis model was chosen to fit the AACW drying kinetic model.

A multiple linear regression analysis of the parameters of the Lewis model shown in Table 5 yields expressions for *k* concerning temperature *T* and relative humidity *RH*:(3)$$k=0.75562+0.00641\;T-2.51294RH,R^{2}=0.92151$$

An AACW drying kinetic model is obtained by substituting *k* into the Lewis model:(4)$$M_{R}=\exp(-(0.75562-0.00641\;T-2.51294RH)t)$$

Finally, the time-varying moisture content model of AACW is obtained by *M_R_* = *ω_t_*_/_*ω*_0_, with the formula as follows:(5)$$\omega_{t}=\omega_{0}\exp(-(0.75562-0.00641T-2.51294RH)t)$$

In this formula, *ω**_t_*** is the moisture content at time *t*; *ω*_0_ initial moisture content; *T* is the temperature (°C); *RH* is the relative humidity; *t* is the time (days).

## 4. Prediction Model of the Equilibrium Moisture Content of AACW

The average temperature, relative humidity, and the equilibrium moisture content of the AACW specimens under different test environments are shown in Table 7.

The equilibrium moisture content of wallboard was related to the temperature and relative humidity of the environment in which they were located. There have been some semi-theoretical and semi-empirical equilibrium moisture content prediction models: the Henderson model [14], modified-Henderson model [15,16,17], Chung–Pfost model [18], modified-Halsey model [19], modified-Oswin model [20], etc. In this paper, the modified-Oswin model, which is suitable for modeling porous solid materials, was used to fit the equilibrium moisture content prediction model of AACW. The modified-Oswin model is shown in Equation (6):(6)$$\omega_{e0}=(b_{0}\exp(\frac{b_{1}}{T})){\left(\frac{RH}{1-RH}\right)}^{b_{2}}$$

In this formula, *ω_e_*_0_ is the predicted value of the equilibrium moisture content; *b*_0_, *b*_1_, *b*_2_ are parameters; *T* is the temperature (°C); *RH* is relative humidity.

The modified-Oswin model was applied to fit the data in Table 7 through MATLAB software, to obtain the equilibrium moisture content prediction model for AACW, as in Equation (7):(7)$$\omega_{e0}=(0.0022\exp(\frac{96.9739}{T})){\left(\frac{RH}{1-RH}\right)}^{0.0917}$$

The determination coefficient *R*^2^ fitted by Equation (7) is 0.99, which was used to predict the balanced moisture content of AACW in different environments.

## 5. Changes in the Drying Shrinkage Value of AACW with the Moisture Content of AACW

### 5.1. Test of the Drying Shrinkage Performance of AACW

Two initial moisture contents were set up for the test: the moisture content in a saturated condition, and the moisture content one day after leaving the kettle, to evaluate the relationship between the drying and shrinkage properties of wallboard and initial moisture content. In order to explore the relationship between ambient temperature, relative humidity, and the drying and shrinkage performance of wallboard, two maintenance environments were set up: a natural maintenance environment, and a standard maintenance environment. The temperature and relative humidity in the standard curing environment were kept constant, and the temperature and relative humidity in the natural curing environment were constantly changed. In this test, the drying shrinkage value of the wallboard specimens was measured from the initial moisture content loss to the equilibrium moisture content 10 days later.

The same manufacturer provided the AACW at equilibrium moisture content for the test. Twelve specimens with dimensions of 400 × 100 × 100 mm were taken from the wallboard material one day after discharge from the kettle. Half of the AACW specimens were immersed in water until saturation and then placed in the standard curing environment (ambient temperature 20 ± 1 °C, relative humidity 90%) and natural curing environment, respectively. The other half of the specimens were placed directly in the standard and natural curing environments, respectively, as shown in Table 8. The relative humidity and temperature at 10 a.m. daily in the natural curing environment were recorded. The moisture content of the specimens was measured every day, and the average value of the three specimens under a specific condition of curing environment and initial moisture content was taken as the standard value of moisture content of the wallboard. The drying shrinkage value of the specimen was measured, taking the average value of the three specimens under the condition of a specific maintenance environment and initial moisture content as the expected value of shrinkage of the wallboard.

In Table 8 showing Z-S(N)-1(2), Z represents the AACW test, S represents the standard maintenance environment, N represents the maintenance environment and the natural maintenance; the numbers 1(2) mean the initial moisture content of specimens under different tests: 1 is the unsubmerged test, and 2 is the submerged saturated test.

The temperature and relative humidity changes in the natural curing environment during the test are shown in Figure 4. The mean temperature during the test period was 22.5 °C, and the average relative humidity was 80%.

The specific test steps were as follows:(1)Select twelve surface-defect-free specimens without honeycombing, cracks, and pores from the autoclaved aerated concrete plate one day after leaving the kettle. The test piece numbers were Z-S-1, Z-S-2, Z-N-1, Z-N-2. Install the shrinking head on the specimen.(2)No. Z-S-2 and Z-N-2 specimens were immersed in water at 20 ± 2 °C, with the top of the specimen 20 mm above the water surface for 72 h. No. Z-S-1 and Z-N-1 specimens were directly immersed in step (3).(3)Measure the initial length *L*_0_ of each test block, accurate to 0.01 mm, and weigh the mass of the test block, accurate to 0.01 kg.(4)The specimens numbered Z-S-1 and Z-S-2 were placed in a standard conditioning box with a temperature of 20 ± 1 °C and relative humidity of 90%. The specimens numbered Z-N-1 and Z-N-2 were placed in a natural curing environment. The specimens’ length *L_t_* and mass mt were measured daily in the early stages; after that, the length *L_t_* and mass mt of the specimens were measured every 2~5 d as the shrinkage gradually stabilized. The mass mt of the specimens was measured every day until the specimens reached the equilibrium moisture content and then pushed back for 10 days after the cutoff, during which the temperature T and relative humidity RH in the natural environment were recorded daily at 10 a.m.(5)The specimens were baked in a drying oven at a temperature of 105 ± 5 °C for 24 h. After 24 h, the mass was weighed every 2 h when the difference between the mass of the specimen before and after was not more than 0.2%, the weight m_d_ of the specimen after drying was measured, accurate to 0.01 kg.(6)Moisture content of test piece at time t (*ω_t_*/%):
(8)$$\omega_{e0}=(0.0022\exp(\frac{96.9739}{T})){\left(\frac{RH}{1-RH}\right)}^{0.0917}$$

Take the average moisture content of 3 specimens in the same test environment as the moisture content of the wallboard in this test environment.

(7)Drying shrinkage value of a specimen at moment t *S_t_* (mm/m) [21]:
(9)$$S_{t}=\frac{L_{0}-L_{t}}{L_{0}-(\eta_{1}+\eta_{2})}\times 1000$$

In this formula, (*η*_1_ + *η*_2_) is the sum of the lengths of the two shrink heads (mm).

Take the average value of drying shrinkage of 3 specimens under the same test environment as the drying shrinkage value of the wallboard under this test environment.

### 5.2. Analysis of Test Results

#### 5.2.1. Changes of Moisture Content with Time

From the tests, the change curve between the moisture content and the time of the wallboard test piece with different initial moisture contents and under different curing conditions was obtained, as shown in Figure 5.

The tests also obtained the balanced moisture content and the time from the initial water loss to the balanced moisture content (the balanced moisture content time) of the wallboard test pieces in the two curing environments, as shown in Table 9.

Available from Figure 5 and Table 9:(1)The equilibrium moisture content of the wallboard specimens under standard maintenance was higher than those under natural maintenance. This was because the ambient relative humidity of standard maintenance was significant, and the larger the ambient relative humidity, the higher the equilibrium moisture content of the wallboard specimen, which was consistent with the conclusion of the equilibrium moisture content test.(2)The wallboard specimen with a saturated moisture content needed more than 44 days to reach the equilibrium moisture content under the natural maintenance environment, and the wallboard specimen one day after the kettle was discharged needed more than 37 days to reach the equilibrium moisture content under the natural environment. Since the size of the AACW specimen was much smaller than the actual size of wallboard in engineering applications, the moisture content of the wallboard will change more slowly in actual projects.

#### 5.2.2. Changes in Drying Shrinkage Value of AACW with Time

The drying shrinkage deformation curves of the wallboard specimens with different initial moisture contents under different maintenance conditions were plotted according to the test data, as shown in Figure 6.

As shown in Figure 6:(1)The drying shrinkage value of AACW was closely related to its curing time: along with the increase in curing time, the drying shrinkage value of the wallboard specimen increased continuously, and its drying shrinkage was intense in the early stage, and stabilized in the later stage. From this, it was necessary to ensure the maintenance time of AACW, so that the drying shrinkage value during the maintenance period could be partially controlled, so that the drying shrinkage value during the use period would be less than the drying shrinkage value at the critical cracking state of the wall.(2)The drying shrinkage value of AACW was about 0.092 mm/m for 15 d, after one day out of the kettle, and about 0.235 mm/m when it finally became stable. Therefore, the maintenance age limit of AACW before installation in a wall after removal from the kettle was 15 d, to meet a drying shrinkage value less than the safe drying shrinkage value during use.

The ambient temperature and relative humidity greatly influenced the drying shrinkage deformation of AACW: the higher the ambient temperature and the lower the relative humidity, the greater the drying shrinkage value of the wallboard. In the test, under the premise of comparable initial moisture content, the drying shrinkage value of the wallboard in the natural curing environment was more significant than its drying shrinkage value in the standard curing environment. Therefore, strengthening the maintenance of AACW was an effective measure to reduce its drying shrinkage value and the cracking of the wallboard filling wall.

(3)The drying shrinkage value of AACW specimens was related to their initial moisture content, and the drying shrinkage value of wallboard specimens in saturated moisture content (initial moisture content 68.22% or 70.44%) was greater than that of wallboard specimens for moisture content, one day after discharge from the kettle (initial moisture content 25.88% or 28.50%) in the test, under the conditions of the curing environment. Therefore, strict control of the moisture content of AACW on the wall was beneficial in reducing the drying shrinkage deformation during the use of the wallboard, thus reducing the risk of cracking the wallboard infill walls.

#### 5.2.3. Time-Varying Model of the Drying Shrinkage Value of AACW with Different Initial Moisture Contents

From the experimental data in Figure 6, a drying shrinkage time-varying model of AACW was fitted (Table 10).

In Table 10, *t* is the maintenance time (d); (*t*) is the drying shrinkage value (mm/m) at time *t* in the standard maintenance environment.

A comparison of the drying shrinkage test values of AACW with the time-varying model calculation curves is shown in Figure 7. As seen in Figure 7, the drying shrinkage time variation model of AACW agreed well with the experimental values, and the drying shrinkage values of AACW can be estimated using the model predictions in Table 10.

#### 5.2.4. Change of Drying Shrinkage Value with Moisture Content

According to the test results, the variation law of the accumulated drying shrinkage value of AACW during water loss in natural and standard curing environments was obtained, as shown in Figure 8. The vertical coordinate in the figure is the cumulative drying shrinkage value that occurs when the test block loses water from the initial moisture content to the moisture content corresponding to the horizontal coordinate.

From Figure 8, it can be seen that the cumulative drying shrinkage value of AACW increased with the increase of water loss, from the initial moisture content to equilibrium moisture content. Under a natural curing environment, the moisture content of the specimens varied, and their cumulative drying shrinkage values increased slowly from the initial moisture content loss to a 20% interval. The growth rate of the accumulated drying shrinkage value of the specimen changed from 20% to 10% when the moisture content of the specimen was further reduced. When losing water from 10% to the equilibrium moisture content in another step, the accumulated drying shrinkage value of the specimen increased with the change in its moisture content. Under the standard maintenance environment, the growth law of the accumulated drying shrinkage value when the specimen lost water from the initial moisture content to the equilibrium moisture content was the same as the growth law of the accumulated drying shrinkage value when the specimen lost water from the initial moisture content to 20%, under the natural environment.

We assumed that the allowable drying shrinkage value of the wall for regular use was 0.2 mm/m [15]. From Figure 8a, we can see the optimal and safe moisture content of AACW was 15% when it was installed on the wall under natural maintenance. In this condition, the drying shrinkage value of AACW can achieve the safety requirements. The results allow a reasonable maintenance of AACW, and an effective crack-control method for AACW was found.

## 6. Conclusions

In order to find an effective control method to reduce the cracking problem of AACW. This study used a theoretical model and experimental research to explore the interrelationship between the moisture content and the dry shrinkage of AACW. Five different test environments were set up, to investigate the process of moisture content reduction to the equilibrium moisture content of the immersed saturated specimens. The drying shrinkage values of the specimens were also investigated under different initial moisture contents and different environments. Conclusions were drawn as follows:(1)The pore characteristics of autoclaved aerated concrete enable autoclaved aerated concrete to adsorb more water in the submerged state. Therefore, AACW has high saturated moisture content characteristics. These specimens of AACW with a saturated moisture content need more than 44 days to reach the equilibrium moisture content under a natural maintenance environment. Moreover, AACW one day after kettle discharge still needs more than 37 days to reach the equilibrium moisture content under a natural environment.(2)A time-varying moisture content model and equilibrium moisture content prediction model of AACW were obtained from the experimental study. According to the conditions and the hypothesis of the test, the moisture content of AACW decreased rapidly in the early stages and then decreased gradually to zero in the later stage, and finally, the specimens reached the equilibrium moisture content. The higher the ambient temperature and lower the relative humidity, the lower the equilibrium moisture content of AACW and the less time it took to dry, from a saturated moisture content to an equilibrium moisture content. The equilibrium moisture content of the wallboard under standard maintenance was higher than that under natural maintenance.(3)A prediction model of the drying shrinkage of AACW was obtained by fitting the drying shrinkage curve of the AACW based on the change of drying shrinkage values with time. The results show that the drying shrinkage value of AACW increases with time, and the drying shrinkage speed is fast in the early stages, and tends to be stable in the later stages. In addition, the drying shrinkage value of AACW increased with the initial moisture content and the ambient temperature, but decreased with the relative humidity.(4)When the AACW lost water, from the initial moisture content to the equilibrium moisture content, the accumulated dry shrinkage value of the wallboard specimens increased with their water loss. According to the test results and the hypothesis of the test, the best time for installing AACW in a wall is 15 d after discharge from the kettle. Controlling the moisture content of AACW in the wall can achieve a best value of 15%. It is favorable to reduce the drying shrinkage deformation during the use of AACW, and the controlling method can meet the requirement that the drying shrinkage value during use is less than the safe drying shrinkage value of 0.2 mm/m. These measures from this research can reduce the risk of cracking of AACW.

The results provide a scientific basis for the reasonable maintenance and profitable control of drying shrinkage cracking of AACW.

## Figures and Tables

**Figure 1 materials-15-05582-f001:**
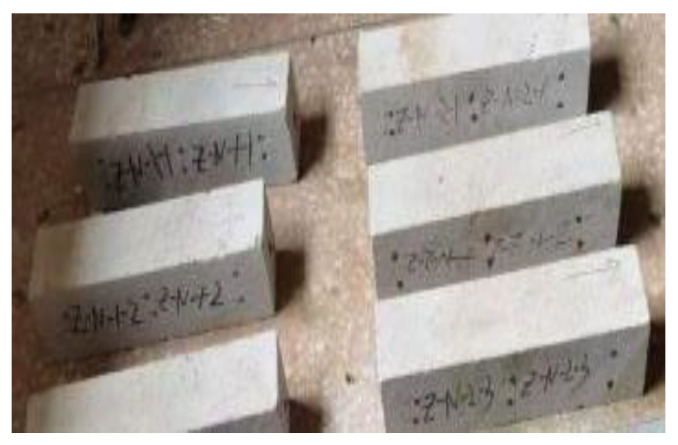
Autoclaved aerated concrete wallboard specimens.

**Figure 2 materials-15-05582-f002:**
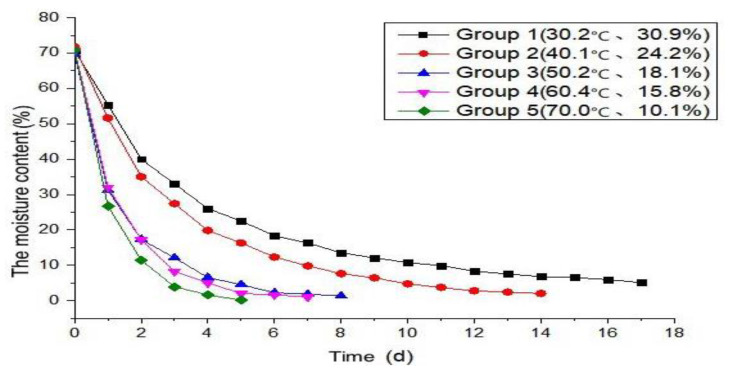
Moisture content of five specimens with time under different test environments.

**Figure 3 materials-15-05582-f003:**
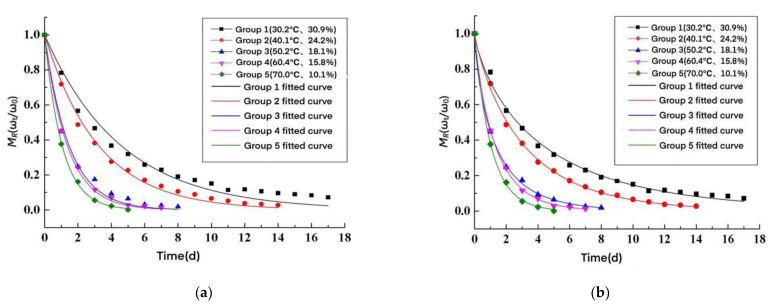
Comparison between the measured values and the theoretical model. (**a**) The Lewis model, (**b**) Page model, (**c**) Henderson–Pabis model, and (**d**) Wang–Singh model.

**Figure 4 materials-15-05582-f004:**
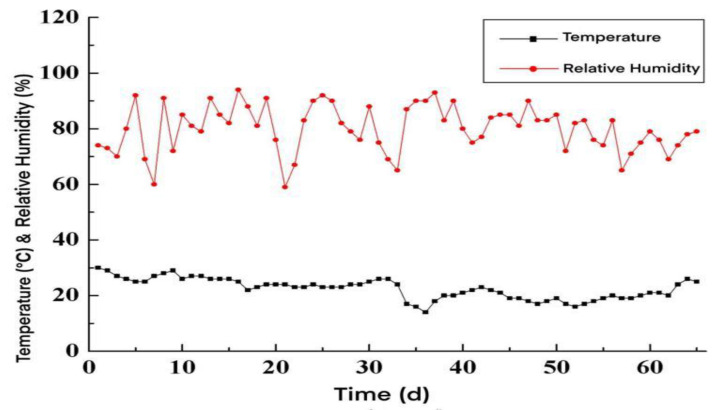
The change in temperature and relative humidity in the natural curing environment.

**Figure 5 materials-15-05582-f005:**
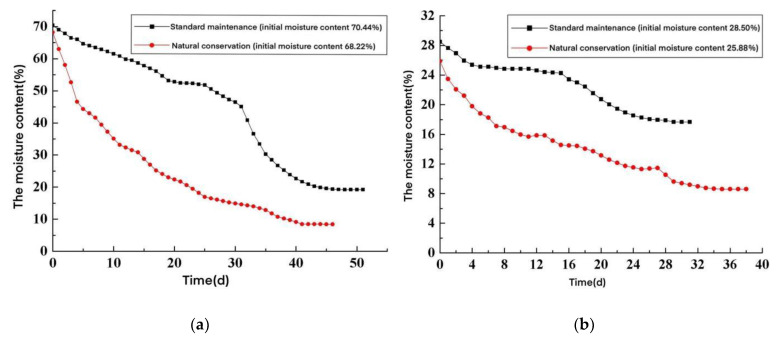
Curve of test moisture content with time. (**a**) Saturated moisture content state and (**b**) moisture content after one day out of the kettle.

**Figure 6 materials-15-05582-f006:**
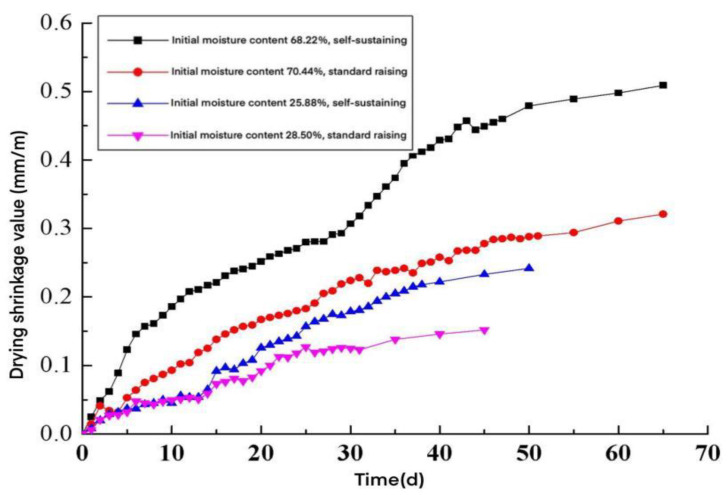
The curve of drying shrinkage value with time.

**Figure 7 materials-15-05582-f007:**
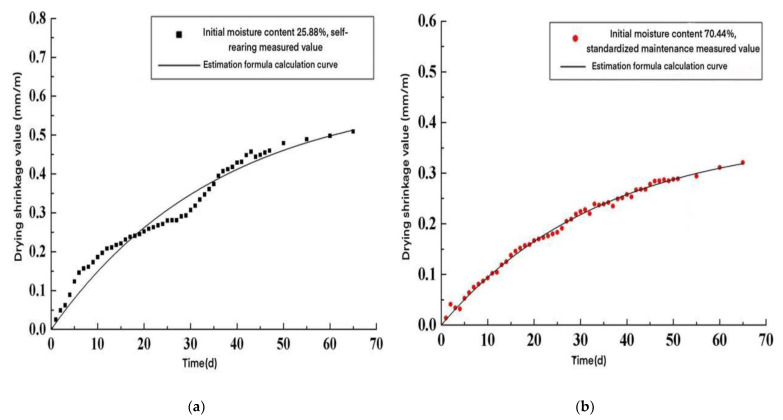
Comparison of the specimen drying shrinkage values and the time-varying model calculation curve. (**a**) Initial moisture content 68.22%, self-sustaining; (**b**) initial moisture content 70.44%, standard raising; (**c**) initial moisture content 25.88%, self-sustaining; and (**d**) initial moisture content 28.50%, standard raising.

**Figure 8 materials-15-05582-f008:**
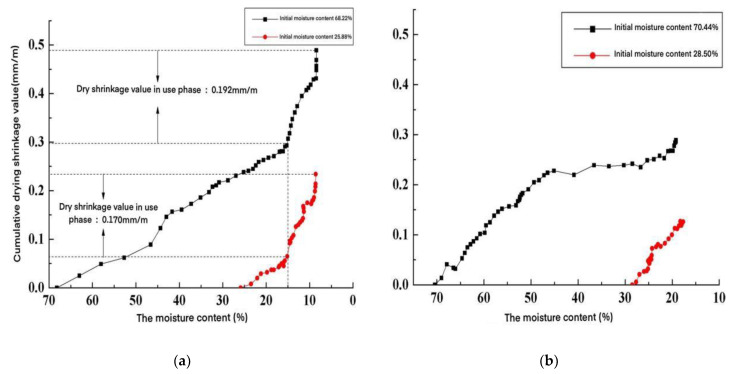
Change pattern of the cumulative drying shrinkage value during tests of moisture content loss. (**a**) Natural maintenance, and (**b**) standard maintenance.

**Table 1 materials-15-05582-t001:** Equilibrium moisture content test program.

Group	The Specimen Number	Temperature/°C	Number of Specimens
1	Z-30-1~Z-30-3	30	3
2	Z-40-1~Z-40-3	40	3
3	Z-50-1~Z-50-3	50	3
4	Z-60-1~Z-60-3	60	3

**Table 2 materials-15-05582-t002:** Daily temperature and relative humidity in the box.

Group	Project	Test Day																	Average Value	Coefficient of Variation/%
		1 d	2 d	3 d	4 d	5 d	6 d	7 d	8 d	9 d	10 d	11 d	12 d	13 d	14 d	15 d	16 d	17 d		
1	*T* (°C)	30.4	31.0	30.2	29.9	29.8	29.8	30.4	30.7	30.8	30.2	29.6	29.4	29.8	30.4	30.5	30.1	30.8	30.2	8.5
	*RH* (%)	31.2	32.4	30.2	30.9	31.0	31.2	31.8	31.2	30.7	30.1	30.4	30.5	31.0	31.1	30.7	30.5	30.9	30.9	1.0
2	*T* (°C)	40.2	40.9	40.6	40.7	39.9	39.4	40.8	40.2	40.3	39.4	39.7	38.9	40.2	40.7	—	—	—	40.1	3.3
	*RH* (%)	23.9	23.8	23.9	24.7	24.5	24.1	24.6	24.2	24.1	23.8	23.9	23.4	24.9	24.7	—	—	—	24.2	1.8
3	*T* (°C)	50.4	50.6	49.8	49.5	49.6	50.8	50.2	50.3	—	—	—	—	—	—	—	—	—	50.2	0.9
	*RH* (%)	18.2	17.9	17.8	18.6	18.4	17.3	19.1	17.4	—	—	—	—	—	—	—	—	—	18.1	3.4
4	*T* (°C)	60.7	60.4	59.8	59.6	60.9	61.0	60.2	—	—	—	—	—	—	—	—	—	—	60.4	0.9
	*RH* (%)	15.9	16.5	15.8	15.4	16.9	15.2	15.1	—	—	—	—	—	—	—	—	—	—	15.8	4.3
5	*T* (°C)	70.4	70.8	69.2	69.5	70.2	—	—	—	—	—	—	—	—	—	—	—	—	70.0	0.9
	*RH* (%)	10.5	10.4	9.8	9.6	10.2	—	—	—	—	—	—	—	—	—	—	—	—	10.1	3.8

**Table 3 materials-15-05582-t003:** Daily moisture content values of specimens of five groups (%).

Time/d	Group 1	Group 2	Group 3	Group 4	Group 5
0	70.7	71.8	69.6	70.7	70.8
1	55.4	51.6	31.2	32	26.7
2	40.0	35.0	17.3	17.2	11.4
3	33.0	27.4	12.1	8.2	3.9
4	26.0	19.8	6.5	5.0	1.6
5	22.5	16.3	4.5	2.0	0.08
6	18.3	12.3	2.2	1.6	—
7	16.3	9.8	1.8	1.0	—
8	13.5	7.6	1.3	—	—
9	12.1	6.4	—	—	—
10	10.7	4.7	—	—	—
11	9.9	3.7	—	—	—
12	8.3	2.7	—	—	—
13	7.5	2.4	—	—	—
14	6.8	2	—	—	—
15	6.5	—	—	—	—
16	5.9	—	—	—	—
17	5.1	—	—	—	—

**Table 4 materials-15-05582-t004:** Mathematical model of drying dynamics.

Model Name	Model	Remarks
Lewis	$M_{R}=\exp(-kt)$	The *k*, *N*, *a*, and *b* are the respective model coefficients, *M_R_* is moisture ratio, *M_R_* = *ω_t_*_/_*ω*_0_, *ω_t_* is the moisture content at time *t*, *ω*_0_ is the initial moisture content.
Page	$M_{R}=\exp(-kt^{N})$	
Henderson-Pabis	$M_{R}=a\exp(-kt)$	
Wang-Singh	$M_{R}=1+at+bt^{2}$	

**Table 5 materials-15-05582-t005:** Calculation parameters of the drying kinetic model.

Group	Environmental Temperature and Humidity	Lewis	Page		Henderson–Pabis		Wang–Singh	
		*k*	*k*	*N*	*a*	*k*	*a*	*b*
Group 1	30.2 °C, 30.9%	0.21830	0.31770	0.78113	0.94011	0.20364	−0.15440	0.00624
Group 2	40.1 °C, 24.2%	0.30895	0.36741	0.87611	0.97383	0.30054	−0.19667	0.00960
Group 3	50.2 °C, 18.1%	0.68162	0.80036	0.76846	0.97890	0.66747	−0.36824	0.03226
Group 4	60.4 °C, 15.8%	0.73433	0.78426	0.89256	0.99210	0.72915	−0.41585	0.04103
Group 5	70.0 °C, 10.1%	0.95391	0.96590	0.96897	0.99861	0.95287	−0.07415	0.07415

**Table 6 materials-15-05582-t006:** Model-fit determinable coefficient (*R*^2^).

Group	Lewis	Page	Henderson–Pabis	Wang–Singh
Group 1	0.97396	0.99460	0.97758	0.91168
Group 2	0.99408	0.99856	0.99452	0.91650
Group 3	0.99038	0.99926	0.98966	0.86452
Group 4	0.99822	0.99961	0.99802	0.91441
Group 5	0.99961	0.99961	0.99952	0.94552

**Table 7 materials-15-05582-t007:** The equilibrium moisture content of the specimens under different test environments.

	Group	1	2	3	4	5
Project						
Average temperature/°C		30.2	40.1	50.2	60.4	70.0
Average relative humidity/%		30.9	24.2	18.1	15.8	10.1
Equilibrium moisture content/%		5.1	2.0	1.3	1.0	0.8
Equilibrium moisture content coefficient of variation/%		13.2	9.2	6.8	12.7	5.7

**Table 8 materials-15-05582-t008:** Test program of drying shrinkage performance.

Test Block	Group	Initial Moisture Content/%	Ambient Condition	Number
Z-S-1	1	28.50	standard curing	3
Z-S-2	2	70.44	standard curing	3
Z-N-1	3	25.88	natural curing	3
Z-N-2	4	68.22	natural curing	3

**Table 9 materials-15-05582-t009:** Changes of moisture content on different condition.

Initial Moisture Content	Saturation Moisture Content		Moisture Content after a Day out of the Kettle	
Curing condition	standard curing	natural curing	standard curing	natural curing
Initial moisture content/%	70.44	68.22	28.50	25.88
Equilibrium moisture content/%	19.20	8.44	17.67	8.61
Balance of moisture content time/d	49	44	29	37

**Table 10 materials-15-05582-t010:** Time-varying model of drying shrinkage value.

Initial Moisture Content/%	Maintenance Environment	Time-Varying Model	Coefficient of Determination
68.22	natural curing	$\varepsilon (t)=0.61287(1-e^{-0.02785t})$	0.96217
70.44	standard curing	$\varepsilon (t)=0.37632(1-e^{-0.02889t})$	0.99586
25.88	natural curing	$\varepsilon (t)=0.30756(1-e^{-0.02780t})$	0.95422
28.50	standard curing	$\varepsilon (t)=0.22665(1-e^{-0.02692t})$	0.96838

## Data Availability

The data presented in this study are available on request from the corresponding author.

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
