# Peer review of "Experimental Study on the Interrelationship between the Moisture Content and Drying Shrinkage of Autoclaved Aerated Concrete Wallboard"

_materials, 2022, doi:10.3390/ma15165582_

Round 1

Reviewer 1 Report

It is required to check the writing of the word "Table" as it was written as a table in different places. It is necessary to check the spelling of the first sentence in the conclusion. Also, it is required to unify the format of the references 

Author Response

Response to Reviewer 1 Comments

Dear professor,

Nice to receive your report, we are grateful for your comments and suggestions. And our response as follows:

Comment 1: It is required to check the writing of the word "Table" as it was written as a table in different places.

Response 1: The writing of the word "table" in different places were replaced with "Table", and the detailed description of each "Table" words were added and revised in the paper. Moreover, the terminology throughout the text was unified and missing numbers were filled.

Comment 2: It is necessary to check the spelling of the first sentence in the conclusion.

Response 2: The spelling of the first sentence in the conclusion was checked and revised, and the
grammatical errors were checked.

Comment 3: Also, it is required to unify the format of the references.

Response 3: The the format of the references has been adjusted in the paper.

Reviewer 2 Report

The experimental investigation on the Effect about the Moisture Content of Autoclaved Aerated Concrete Wallboard on Drying Shrinkage reported in this study is an interesting topic. The Introduction section provides sufficient background of this study and included sufficient literature. The references cited are appropriate to this research. The research is design appropriate. The methods of testing are elaborately described. The results are clearly presented with the help of charts and tables. The conclusions section is supported by the results obtained from the study.

However, the following corrections needs to be carriedout before the acceptance of this manuscript,

1.    Elaborate the research need

2.    Check the figure number 3. Figure 2 is missing.

3.    Increase the legend size for the figures. It is not clear.

4.    In page 5, it is stated that “its SEM image shows that most of its pores are through pores, and the macro pores are in the shape of ink bottle with "small mouth and big belly". But there is no SEM images in the manuscript.

5.    You have cited the references in number format, through out the manuscript but in REFERENCE section it is in different format. there are issues regarding the reference style. Follow the journal guidelines strictly.

6.    The citations of the references and the references in the list shall be cross checked for consistency. Some of the citations are not included in the references and some references in the list are not cited in the text.

Author Response

Response to Reviewer 2 Comments

Dear professor,

Nice to receive your report, we are grateful for your comments and suggestions. And each items are responded and explained as follows:

Comment 1: Elaborate the research need.

Response 1: The introduction section has been revised, and the research need was elaborated.

Comment 2: Check the figure number 3. Figure 2 is missing.

Response 2: The typing errors were revised, the “Figure 3” in the fifth page was replaced by “Figure 2”.

Comment 3: Increase the legend size for the figures. It is not clear.

Response 3: The scale of the figures was enlarged. And the legend size was increased for the figures.

Comment 4: In page 5, it is stated that “its SEM image shows that most of its pores are through pores, and the macro pores are in the shape of ink bottle with "small mouth and big belly". But there is no SEM images in the manuscript. 

Response 4: In page 5, corresponding reference was added to explain this comment.

Comment 5: You have cited the references in number format, through out the manuscript but in REFERENCE section it is in different format. there are issues regarding the reference style. Follow the journal guidelines strictly.

Response 5: Following the journal guidelines strictly, the references format were revised. In the article, we have cited the references in number format. In REFERENCE section, it is in same format.

Comment 6: The citations of the references and the references in the list were cross checked for consistency. Some of the citations are not included in the references and some references in the list are not cited in the text.

Response 6: The citations of the references and the references in the list were cross checked for consistency. In addition, the original reference “14” was removed and some new references were added.

Reviewer 3 Report

Reviewers' comments:

Manuscript number: materials-1849777

Title: Experimental Study on the Effect about the Moisture Content of Autoclaved Aerated Concrete Wallboard on Drying Shrinkage.

Comments: 

The manuscript reported on Experimental Study on the Effect about the Moisture Content of Autoclaved Aerated Concrete Wallboard on Drying Shrinkage. The manuscript needs a detailed editing. It cannot be recommended for publication in the present form. I hope the following points would be helpful for the authors.

- Qualitative information’s are missing in abstract.

- Add more suitable keywords.

- In the introduction section, write the novelty of the work and the problem statement clearly. Authors fails to explain the novelty and importance of the proposed research work thus substantial discussion is essential.

- Please provides the references for all equations and formula.

- Figure 3, not clear make clear.

- 3. The time-varying model of the moisture content – should be improve.

- Author should check Figure numbers.

- 5.2.4. Changes of Drying Shrinkage Value with Moisture Content – should be improve.

- The conclusion is too long, conclusion should be concise.

- References: author should use reference number.

- References: there are recent references in 2021 and 2022 treating the same subject, you can use. - Make all references in same format for volume number, page numbers and journal name, because it is difficult to searching and reading.

- Some English and grammar mistakes are present that need to be correct to improve the quality of the manuscript.

Based on these, I advise the authors to rectify the above-mentioned errors and we hope to re-evaluate the revised manuscript.

Author Response

esponse to Reviewer 3 Comments

Dear professor,

Nice to receive your report, we are grateful for your comments and suggestions. And each items are responded and explained as follows:

Comment 1: Qualitative information’s are missing in abstract.

Response 1: Qualitative information’s were added in the abstract.

Comment 2: Add more suitable keywords.

Response 2: More suitable keywords were added in the revised manuscript.

Comment 3: In the introduction section, write the novelty of the work and the problem statement clearly. Authors fails to explain the novelty and importance of the proposed research work thus substantial discussion is essential.

Response 3: The introduction section was revised. Moreover, the novelty of the work and the problem statement were added.

Comment 4: Please provides the references for all equations and formula. 

Response 4: Some references were provided for the mathematical model of drying dynamics and all equations and formula.

Comment 5: Figure 3, not clear make clear.

Response 5: We make clear all Figures, Figure 3 has been changed and all Figures were revised.

Comment 6: 3. The time-varying model of the moisture content – should be improve.

Response 6: The depiction of the time-varying model of the moisture content was improved, and some references were added.

Comment 7: Author should check Figure numbers.

Response 7: All figure numbers were checked, and the Figure 3 of the fifth page was revised.

Comment 8: 5.2.4. Changes of Drying Shrinkage Value with Moisture Content – should be improve.

Response 8: The depiction of Changes of Drying Shrinkage Value with Moisture Content was improved.

Comment 9: The conclusion is too long, conclusion should be concise.

Response 9: The conclusions have been refined, please see the annex for details.

Comment 10: References: author should use reference number.

Response 10: The reference format was changed, and all references use numeric format.

Comment 11: References: there are recent references in 2021 and 2022 treating the same subject, you can use. - Make all references in same format for volume number, page numbers and journal name, because it is difficult to searching and reading.

Response 11: Following the journal guidelines strictly, the references format were revised. In the article, we have cited the references in number format. In REFERENCE section, it is in same format. And, the citations of the references and the references in the list were cross checked for consistency.

Comment 12: Some English and grammar mistakes are present that need to be correct to improve the quality of the manuscript..

Response 12: Some English and grammar mistakes of the manuscript were corrected and the terminology throughout the text was revised.

Reviewer 4 Report

See attached file

Author Response

Response to Reviewer 4 Comments

Dear professor,

Nice to receive your report, we are grateful for your comments and suggestions. And each items are responded and explained as follows:

Comment 1: Author(s) are advised to modify the topic of the MS to make it more meaningful. Like, Experimental Study on interrelationship between the Moisture Content and Drying Shrinkage of Autoclaved Aerated Concrete Wallboard.

Response 1: We are grateful for this comment and suggestion, the topic was modified.

Comment 2: The submitted manuscript needs thorough language editing as the manuscript has many grammatical and typographical errors right from the abstract to conclusions. For example, "However, the easy cracking problem of AACW in the using process becomes more prominently, and it is necessary to figure out the relationship between the moisture content and the dry shrinkage value of AACW." (Abstract).

Response 2: Many grammatical and typographical errors right from the abstract to conclusions were revised.

Comment 3: Abstract should explain clear hypothesis of the research.

Response 3: The hypothesis of the research was explained in Abstract.

Comment 4: Introduction section last paragraph should include the novelty of the present study including the objectives of the study. 

Response 4: The novelty of the present study and the objectives of the study were added in the introduction section.

Comment 5: The meaning of any symbol used should be given first.

Response 5: The meaning of the formula symbols was revised in the text. And the meaning of all symbols were given at the moment of the first appearance time. In addition, the meaning of the symbols in the formula was explained in "The formula".

Comment 6: Author(s) are advised to add some numerical works related to topic.

Response 6: Some numerical works and references related to topic were added.

Comment 7: The structure of MS should be updated. Authors are advised to add a separate experimental methodology section and results and discussion section. The test data and discussion should be given in separate sections.

Response 7: The structure of MS was updated, and "selection and test scheme"; "Test results and Analysis" were given in the manuscript. In addition, the test data and discussion were given in separate sections. In the article, each section has a separate analysis and discussion.

Comment 8: All references given at the end should be numbered and cited in the MS. It is also advised add some more relevant references.Following reference is not cited in the MS and should be removed.J. CHIRIFE, H. A. IGLESIAS. Prediction of the effect of temperature on water sorption isotherms of food material[J]. International Journal of Food Science and Technology. 1976, 11(2):109-116.

Response 8: The original reference "J. CHIRIFE, H. A. IGLESIAS. Prediction of the effect of temperature on water sorption isotherms of food material[J]. International Journal of Food Science and Technology. 1976, 11(2):109-116." was deleted and replace it with "Wang, B.H. A review of drying kinetics research. Drying Technology and Equipment. 2009, 7, 1-6.". In addition, some relevant references were added.

Reviewer 5 Report

Paper entitled “Experimental study on the effect about the moisture content of autoclaved aerated concrete wallboard on drying shrinkage meets the necessary standards for publication in this journal.

Article is well organized and problem is comprehensively reviewed, however, some minor changes are required.

General comments on the text as a whole:

1. I am asking for a general correction of the provided manuscript in terms of editing - too often there are double spaces between words, or there are no spaces between words. In addition, please check the record of the obtained test results and the units used. Generally, in the provided article, the record of test results and units is correct and understandable, but for example on page 9 the form "20mm" was used. I think it is more correct to write "20 mm". Records in this manner can also be found on other pages of the article, e.g. on page 14.

2. Please correct the text editing, e.g. in terms of the titles of figures and tables - the title is not always on the page on which the figure/table is placed.

3. Please correct the numbering of the figures. It seems that Figure 3 is used twice, by mistake, I hope. In addition, Figure 2 is missing from the manuscript, but is cited in the text on pages 4 and 5.

4. Please standardize the record of the headings of figures. Once it is written in plain font, another time it is bold. This applies, for example, to the title of figure 8.

 5. Please consider and improve the literature citations – the main text uses the numbers of individual items of literature, but the list of references does not contain numbers at all. Individual literature items are marked with dots. They should be given numbers.

6. Attention when writing references. They are not unitary. Plesae use MDPI Reference List and Citations Style Guide. 

Author Response

Response to Reviewer 5 Comments

Dear professor,

Nice to receive your report, we are grateful for your comments and suggestions. And each items are responded and explained as follows:

Comment 1: I am asking for a general correction of the provided manuscript in terms of editing - too often there are double spaces between words, or there are no spaces between words. In addition, please check the record of the obtained test results and the units used. Generally, in the provided article, the record of test results and units is correct and understandable, but for example on page 9 the form "20mm" was used. I think it is more correct to write "20 mm". Records in this manner can also be found on other pages of the article, e.g. on page 14.

Response 1: The error of "20mm" has been corrected, and the record of the obtained test results and the units used were revised.

Comment 2: Please correct the text editing, e.g. in terms of the titles of figures and tables - the title is not always on the page on which the figure/table is placed.

Response 2: The text was revised. And the textbook has been edited to place the title and figure/tables on one page.

Comment 3: Please correct the numbering of the figures. It seems that Figure 3 is used twice, by mistake, I hope. In addition, Figure 2 is missing from the manuscript, but is cited in the text on pages 4 and 5.

Response 3: The numbering of the figures was revised. In addition, the Figure 3 of the fifth page was changed to Figure 2.

Comment 4: Please standardize the record of the headings of figures. Once it is written in plain font, another time it is bold. This applies, for example, to the title of figure 8. 

Response 4: The record of the headings of figures was revised, the bold title of Figure 8 was changed to plain font.

Comment 5: Please consider and improve the literature citations – the main text uses the numbers of individual items of literature, but the list of references does not contain numbers at all. Individual literature items are marked with dots. They should be given numbers.

Response 5: Following the journal guidelines strictly, the references format were revised. In the article, we have cited the references in number format. In REFERENCE section, it is in same format. And, the citations of the references and the references in the list were cross checked for consistency.

Comment 6: Attention when writing references. They are not unitary. Plesae use MDPI Reference List and Citations Style Guide.

Response 6: Following the journal guidelines strictly, the references format were revised. And, the citations of the references and the references in the list were cross checked for consistency.

Round 2

Reviewer 3 Report

Reviewers' comments:

The authors revised the manuscript according to the reviewers' comments.

Reviewer 4 Report

Authors have responded to the comments and MS may be accepted.